# Practical Use of Quartz Crystal Microbalance Monitoring in Cartilage Tissue Engineering

**DOI:** 10.3390/jfb13040159

**Published:** 2022-09-21

**Authors:** Jakob Naranda, Matej Bračič, Matjaž Vogrin, Uroš Maver, Teodor Trojner

**Affiliations:** 1Department of Orthopaedics, University Medical Centre Maribor, SI-2000 Maribor, Slovenia; 2Department of Orthopaedics, Faculty of Medicine, University of Maribor, SI-2000 Maribor, Slovenia; 3Laboratory for Characterisation and Processing of Polymers (LCPP), Faculty of Mechanical Engineering, University of Maribor, SI-2000 Maribor, Slovenia; 4Institute of Biomedical Sciences, Faculty of Medicine, University of Maribor, SI-2000 Maribor, Slovenia; 5Department of Pharmacology, Faculty of Medicine, University of Maribor, SI-2000 Maribor, Slovenia

**Keywords:** quartz crystal microbalance (QCM), cartilage tissue engineering, biomaterials, nanotopography, layer-by-layer (LbL), nanofilms

## Abstract

Quartz crystal microbalance (QCM) is a real-time, nanogram-accurate technique for analyzing various processes on biomaterial surfaces. QCM has proven to be an excellent tool in tissue engineering as it can monitor key parameters in developing cellular scaffolds. This review focuses on the use of QCM in the tissue engineering of cartilage. It begins with a brief discussion of biomaterials and the current state of the art in scaffold development for cartilage tissue engineering, followed by a summary of the potential uses of QCM in cartilage tissue engineering. This includes monitoring interactions with extracellular matrix components, adsorption of proteins onto biomaterials, and biomaterial–cell interactions. In the last part of the review, the material selection problem in tissue engineering is highlighted, emphasizing the importance of surface nanotopography, the role of nanofilms, and utilization of QCM as a “screening” tool to improve the material selection process. A step-by-step process for scaffold design is proposed, as well as the fabrication of thin nanofilms in a layer-by-layer manner using QCM. Finally, future trends of QCM application as a “screening” method for 3D printing of cellular scaffolds are envisioned.

## 1. Introduction

In the last few decades, quartz crystal microbalance (QCM) has become a powerful and well-established noninvasive technique for numerous measurements clarifying various aspects of biological materials and their interactions. The QCM is an extremely sensitive, practical, and convenient monitoring tool that relies on the resonant piezo crystal oscillation changes to measure nanogram- to microgram-level changes in mass per unit area [1,2]. Various biological substrates from small molecules to whole cells can be investigated with QCM. In medicine, QCM is a useful technology for diagnosing diseases such as cancer and viral and bacterial infections [3,4]. The extended version of QCM with dissipation monitoring (QCM-D) allows for measurements of dissipative energy arising from the dampening of the crystal’s oscillation by the adsorbed layer of (biological) material, thus providing information on its viscoelastic nature [5,6].

QCM has been successfully applied in a broad range of biomedical applications to monitor and explore various surface interactions, in situ thin-film formations, and layer properties [7]. This includes molecular interactions on material surfaces, cell characterization, cell–biomaterial interactions, DNA and protein detection, protein drug target screening, etc. [8,9]. Understanding the interactions of complex biomolecules with nanomaterial surfaces is of the utmost importance for ensuring biocompatibility and further potential biomedical use of new nanomaterials. The interaction of proteins with nanomaterials and cellular interactions are particularly important for implant development, biosensors, and scaffolds in tissue engineering (TE), as well as regenerative medicine [10]. Thus, high-throughput capacity, experimental flexibility, and nanoscale sensitivity make QCM applicable to investigate complex biological problems in advanced research areas of TE.

The state-of-the-art concept of TE combines biocompatible and biodegradable three-dimensional (3D) biomaterials (scaffolds), providing initial support for the desired cells to attach, proliferate, and form their native extracellular matrix (ECM) over longer periods [11,12]. The outcome of TE relies on appropriate material selection and processing strategies for scaffold preparation that preserves the desired phenotype and allows for functional tissue maturation [13,14].

This review focuses on the potential use of QCM in cartilage TE (CTE). Despite all the advancements of TE in the last decade, CTE still presents a significant challenge. Considering the complex arrangement of collagen fibers and distribution of glycosaminoglycans (GAG) in articular cartilage, which is critical for its biomechanical function, the development of artificial recapitulation of cartilage necessitates the use of increasingly advanced analytical methods [15]. Numerous materials, either natural or synthetic, polymers, hydrogels, and composites, have been used in scaffold preparation for CTE [16]. Recently, composites of naturally degradable biopolymers mimicking cartilage ECM have been exploited due to their excellent biocompatibility, low immunological response, and low cytotoxicity, as well as their excellent capability to promote cell adhesion, proliferation, and regeneration of new tissues [17,18]. Among regenerative strategies, 3D printing is becoming an increasingly common technique to fabricate scaffolds for CTE since it allows for the development of biological and anatomical 3D structures according to specific (defined) needs [19,20]. Despite numerous published studies on successful 3D printing of biomaterials and their combinations, there are still challenges in preparing appropriate “biomaterial inks” and processing these materials into self-supporting devices with tunable mechanics, degradation, and bioactivity [21].

Among these, the production of thin nanofilms can be helpful, especially in investigating biomaterial interaction capabilities/potential to determine appropriate combinations of polymers (either natural or synthetic), additives (e.g., nanofibrous cellulose), and active ingredients (e.g., growth factors) to monitor initial cellular behavior, beneficial for further formulation in the form of 3D structures [22], particularly layer-by-layer (LbL) thin nanofilms where several layers of different biomaterials mimicking the complex ECM composition can be deposited on a nanoscale level [23]. This can serve as a prepreparation design for scaffold development in TE to bypass the immediate 3D printing process on a trial-and-error basis with subsequent time-consuming cultivation and conventional analyses (e.g., molecular methods, immunohistochemistry, etc.). Hence, preparing LbL films of biomaterials followed by real-time QCM cell analysis may provide rapid information about biomaterial suitability for a particular cell type and boost our CTE efficiency.

The purpose of this review article is to provide a summary of QCM as a tool to investigate solid–liquid interactions of scaffold surfaces made of biomaterials with biomolecules and various cellular phenomena, including attachment and proliferation. It especially focuses on the potential use of QCM in CTE. Accordingly, the strategy and design of cell-laden scaffolds are described, and the importance of the correct choice of biomaterials and surface nanotopography is emphasized. In recent years, the number of articles on selecting biomaterials in the field of CTE has grown exponentially. However, there is still no suitable solution for selecting biomaterials that would provide a favorable subsequent cellular response in interaction with different biomaterials. The latter is pivotal for innovations in the biomedical field, especially CTE, since the outcome of tissue regeneration mostly depends on the initial cell–biomaterial contact. Special emphasis is placed on in-situ LbL nanofilm preparation and biomolecular interaction studies by QCM to test biomaterial suitability for CTE scaffolds. For this purpose, we propose a step-by-step solution in scaffold design as shown in Figure 1, including: (1) preprocessing 1 (biomaterial selection, thin-film formation, QCM evaluation); (2) preprocessing 2 (cell–biomaterial interactions); (3) processing (bioink formulation, 3D printing, and scaffold fabrication); (4) postprocessing (in vitro cultivation, scaffold characterization); and (5) application (animal models, clinics). Finally, future trends are envisioned in using QCM as a “screening” method for 3D printing cellular scaffolds. According to the authors’ knowledge, this is the first review article to address the application of QCM technology in CTE.

## 2. Materials and Methods

A literature review was conducted on some of the most extensive medical literature databases (PubMed, ScienceDirect, Springer Nature) to obtain studies related to scaffold design and material selection in cartilage tissue engineering. The employed search terms in the form of keywords were “cartilage tissue engineering (CTE)”, “Quartz Crystal Microbalance with Dissipation Monitoring (QCM-D)”, “layer-by-layer (LbL) thin nanofilms”, and “surface nanotopography”. The used Medical Subject Headings (MeSH) identifiers were “articular cartilage”, “tissue engineering”, “regenerative medicine”, “tissue scaffolds”, “biocompatible materials”, “biomimetic materials”, “adsorption”, “cell physiological phenomena”, “cell adhesion”, “cell differentiation”, “quartz crystal microbalance techniques”. With the help of this search algorithm and specific filters (5-year, review), we could find relevant new impactful studies on material selection for a scaffold design in CTE, which were included in this review.

## 3. Sensing Principles of QCM

QCM is a transient-mode resonator that measures the electric properties (frequency, energy dissipation, and half-bandwidth) of an oscillating piezoelectric quartz crystal. The fundamental frequency (*f*_0_) of the crystal depends on the wave velocity (*v_q_*) and the crystal’s thickness (*d*) as *f*_0_ = *v_q_*/2*d*. Its electrical properties are sensitive to changes on its surface and will change when a load, e.g., fluid phase, rigid film, viscoelastic film, is introduced to the crystal’s surface [24]. The change is related to the physical properties of the load, such as density, viscosity, film thickness and mass, and viscoelasticity [24,25]. Models to correlate the physical properties of the load and the electrical properties of the crystal are plenty, and several sensing mechanisms of QCM have been derived from that.

The first correlations were described by Sauerbrey, who found that if a load on the oscillating crystal is uniform, rigid, thin enough (much lower than the decay length of the shear wave), and behaves elastically like the crystal, then added thickness of the load can be treated the same as the oscillation of a thicker crystal [26]. He further derived a relation between the change in mass Δ*m* (added load) with the change in frequency (Δ*f*) as Δm=−C·Δf, where *C* is a mass sensitivity constant defined as $C=\sqrt{\rho_{q}\upsilon_{q}}/2f_{0}^{2}$ (density (*ρ_q_*), wave velocity (*v_q_*), and fundamental resonance frequency (*f*_0_). This groundbreaking relation was fundamental in the determination of adsorbed mass of molecules on solid surfaces from gaseous phase by QCM. However, it is linear only if the crystal’s elastic behavior meets the above-mentioned criteria. As soon as the physical properties, e.g., viscoelasticity of the load, become a significant part of the quartz crystal’s response, the Sauerbrey relation is invalid. Despite that, it is still valuable in limited cases for adsorption of molecules from liquid phase and used regularly in the field of biomaterials. The nonlinear behavior is typical for viscoelastic films in liquid phase. Therefore, more complex models to determine the mass of the load were developed. For example, a more detailed relationship between the oscillating frequency and the physical parameters of a liquid phase was given by Kanazawa and Gordon by including the density and viscosity of the liquid to the relationship [27]. Further advanced models, such as the four-layer theory, additionally compensate for the wettability-driven slip at the solid–liquid interface that perturbs the acoustic wave propagation and modifies the sensor signal [28]. These improvements provide more realistic models of the solid–liquid systems and establish a new QCM sensing mechanism that enables determination of additional physical properties of the load, e.g., density and viscosity [29], and the identification of wetting behavior of the surfaces. Recently, surface topography and surface chemistry have been shown to strongly influence QCM response by altering the wetting behavior on the crystal’s surface [30,31]. Superhydrophobic micropillar topography on QCM surfaces show a significant increase in the sensitivity of the QCM (up to 10-fold) by taking advantages of the coupled resonance between micropillars and the quartz crystal by forming a two-degrees-of-freedom coupled vibration, possessing a coupled resonant frequency and vibration characteristic near the critical pillar height [30,32]. Other studies have found that a hydrophobic surface with water penetrating inside the layer caused a frequency decrease and no changes in dissipation (Sauerbray mass load), while a nonpenetrating hydrophobic surface (contact angle approaching 180°) produced very low changes in frequency and dissipation, which is attributed to the crystal’s resonance remaining defined by the strong reflection from the upper crystal surface, which remains mainly in contact with air despite the immersion of the crystal in the liquid [33]. The thin air film or “plastron” formed on the solid–liquid interface causes decoupling of the crystal’s response. However, this new QCM sensing mechanism offers an opportunity to follow various binding events, including cell attachments, by detecting changes in the surface free energy of the sensitive surface, altering its wettability, and subsequently the QCM response [34]. This sensing mechanism provides the opportunity to gain critical knowledge for designing high-sensitivity QCM biosensing systems and the evaluation of superhydrophobic and superhydrophilic surfaces for a variety of applications [30]. This is extremely important in CTE as cells respond to the nanostructured surface by altering adhesion, organizing the cytoskeleton, and expressing the desired phenotype [35]. Basic information utilized from the sensing mechanisms based on QCM frequency measurements are shown in Figure 2a. Information about the adsorbed (decreasing frequency) and desorbed (increasing frequency) mass of molecules in real time can be obtained by QCM, as well as the kinetics of the adsorption and desorption processes (slope of the frequency change). It is noteworthy that as the crystal’s oscillation propagates into the liquid, its amplitude decays to 1/e of its original amplitude within a certain distance (decay length). The decay length defines the measurable interfacial layer thickness [29]. For a AT-cut quartz crystal with a fundamental frequency of 5 Hz, that would be roughly 250 nm [31]. The crystals oscillate at their fundamental frequency and various overtones (harmonics) at different amplitudes having specific decay lengths. Utilizing this presents an addition to the sensing mechanisms that enables gathering of physical information about the load at various thickness levels, close to the solid–load interface or far at the load–bulk interface. Practically, very accurate film thickness, modulus, and phase angle of a thermoresponsive hydrogel undergoing large swelling changes were calculated in situ by this approach [36].

QCM devices with dissipation monitoring (QCM-D) can also monitor the half-bandwidth variation Δ*Γ* (or the energy dissipation change Δ*D*). Friction of the load above the crystal dampens the oscillation amplitude, producing energy loss (dissipation). Dissipation is defined as *D* = *E*_dissipated_/2π*E*_stored,_ where *E*_dissipated_ is the energy dissipated during one oscillation period and *E*_stored_ is the energy stored during oscillation [37]. The energy loss can be measured by impedance spectroscopy [38], the decay time of the oscillation [39], or the resistance [31]. Thin and rigid loads oscillate with the crystal and cause low energy dissipation, and thick and viscoelastic loads adopt their own oscillation, causing higher energy dissipation. This additional sensing mechanism based on dissipation measurement provides information about the viscoelastic properties of the load (Figure 2b), as rigid or soft layers produce a low or high change in dissipation, respectively. Furthermore, the conformation of the adsorbed molecules can be obtained when combining the frequency and dissipation changes (Figure 2c). Low frequency–high dissipation curves are often produced by flat and loosely bound molecules, while high frequency–low dissipation curves reflect elongated and densely bound molecules. The latter is a very strong QCM tool for TE as it can provide insights into surface mechanisms of biomolecules, e.g., specific cellular behavior such as cell adhesion to solid surfaces, cytotoxicity, or cytoskeleton modifications [40].

### Practical Use of QCM Sensing

QCM is the most popular and widely used acoustic transducer for sensor applications that has become widely acceptable in chemical and biosensing fields. The practical importance of QCM has been well-emphasized in the literature, as this technology can be used in various areas, such as: environmental monitoring [41], food industries [42], viscosity measurement [43], humidity and gas sensors, metal film deposition, biosensing, etc. [44]. The applications of QCM in liquid viscosity measurements are still popular since these sensors can operate in harsh environments such as extreme temperature, acidity, and pressure [43]. Recent technological innovations of the QCM enable the quality assessment of selected food products and the analysis of food raw materials and ingredients for foodborne pathogen detection [42]. There are trends towards increasing use of QCM as a biosensing system based on label-free and real-time biorecognition mechanism. Furthermore, QCM has the ability to detect a broad variety of biomolecules, such as proteins, nucleic acids, peptides, oligonucleotides, hormones, etc. [45]. In particular, the rapid detection process and the high sensitivity of QCM systems make them attractive for the development of novel diagnostic tools. Hence, QCM has gained significant interest in pathogen detection of infectious diseases [3]. Various viruses have been detected via QCM [46,47]. For example, it was shown that the results of QCM-D detection of influenza A and B viruses were comparable in sensitivity and specificity to cell culture methods [48]. Moreover, the QCM biosensor was also applied in cancer detection, e.g., breast cancer cells with high metastatic potential were detected with the help of a QCM biosensor functioning via transferrin receptor interactions [49]. There are also reports where QCM was able to detect other pathogenic and abnormal biomarkers in diseases such as lung, cervical and prostate cancers, Burkitt’s lymphoma, tuberculosis, etc. [50,51,52,53,54].

The growing number of reported studies in which QCM sensors are based on the molecular imprinting technique shows that these sensor systems are promising for selective recognition [2]. In addition, QCM was demonstrated as a reliable, low-cost, highly sensitive, small-sized, rapid-to-use, and portable biosensing tool with a short detection time that offers improvement in early and accurate detection of various biological markers. Therefore, QCM could facilitate the disease treatment process [3] in early detection and quantification of disease-related biomarkers in blood, urine, or saliva. There have been several examples demonstrating the feasibility of QCM-D sensors for the detection of biomacromolecules related to specific clinical diagnosis such as different types of cancers and chronic diseases [50,51,52,53,54]. In one example, QCM-D sensors were used for the detection of glucose, which demonstrated the feasibility of QCM-D sensors for clinical work related to glucose monitoring [55]. In another study, QCM-based biosensors showed a strong potential for in situ quantification of human urine. The developed QCM sensors were endowed with the ability of recognizing subtle differences in the urine’s biochemistry at minimized noise levels [56]. In addition, QCM-D was utilized for rapid automated blood group analysis [57] and for monitoring coagulation homeostasis [58]. The results demonstrated the suitability of the QCM-D-based Clauss fibrinogen assay (CFA) in comparison with common coagulation reference methods [59]. Moreover, a new approach for human seminal fluid analysis using soot-coated QCMs was presented [60]. A QCM-based biosensor was developed for in situ assessment of male gamete quality, incorporating a superhydrophobic soot coating as an interface sensing material. The obtained results reveal the strong potential of the superhydrophobic QCM for future inclusion in diverse laboratory analyses of in vitro fertilization procedures [61]. Overall, the collected literature highlighted the role of the QCM system as a new and nonconventional method for use in clinical medicine to detect various biochemical markers for early diagnosis and to monitor the progress of the disease. Finally, these new bioreceptor and biomarkers based on QCM technology could make an ideal candidate for cheap point-of-care (POC) testing for early detection of diseases [54].

Another important area where the use of QCM-D has become increasingly popular is the investigations of bacterial adhesion, bacterial biofilm formation, antifouling strategies, and antibacterial coatings [62,63,64]. The utility of QCM-D is based on the ability to probe binding and interactions under dynamic conditions in real time. Bacterial adhesion is an important first step in the formation of biofilms that are responsible for persistent infections, raising serious concerns in medical implants [65]. In one study, a model for attachment of *Pseudomonas aeruginosa* was developed using QCM-D [66]. Moreover, the detailed biofilm development stages (adhesion, maturation, and dispersion) of *Escherichia coli* on gold and titanium surfaces was monitored with a QCM device [67]. Apart from this, QCM was also used to study bacteria antibioadhesion behavior by applying various coatings [68,69,70]. For example, it was shown that the soot coatings inhibit the proliferation of *Pseudomonas* species and reduce their quantity compared to an uncoated glass slide [71]. Similarly, silver nanoparticles as antibacterial coatings were shown to possess antimicrobial properties in Gram-negative bacterial strains, which makes them very interesting for a number of practical applications, e.g., prevention of microbial colonization on various medical devices and implants [68]. In general, recent antifouling strategies suggest the employment of low-adhesive materials that can hinder biocontamination [71,72].

Recent application of QCM in whole-cell studies has begun to attract attention. QCM can directly monitor the attachment and spreading of cells on the surface of the quartz sensor crystal and provide the assessment of cellular behavior. In addition, QCM enables the monitoring of interactions with extracellular matrix components, adsorption of proteins onto biomaterials, biomaterial–cell interactions, and functional responses of the cells on the surface of biomaterials [40,73]. The ability to monitor changes of different biomolecules along with the ability to study various cellular phenomena (e.g., adhesion and proliferation) provides an excellent tool for tissue engineering studies. The detailed explanation of the practical use of QCM in tissue engineering is presented in the next chapters with a focus on cartilage TE.

## 4. Potential Use of QCM in Cartilage Tissue Engineering (CTE)

The main purpose of CTE is tissue regeneration, multiplication, and differentiation of cells into a desired tissue-specific form by selecting suitable cellular scaffolds and the cells’ favorable growth conditions [74]. Currently, CTE represents a major challenge due to the specific properties of cartilage tissue, slow growth, characteristic structure of ECM, extraordinary mechanical properties, and a high degree of dedifferentiation in cell cultures [13]. Cellular scaffolds used in CTE are essential because they allow for cultivation in 3D structures and stimulate hyaline cartilage formation. These scaffolds should meet the relevant requirements, including biocompatibility, suitable degradability, appropriate physicochemical, biological, and architectural features (porosity, pore permeability, mechanical properties, etc.), and stimulate the cartilage phenotype [75]. Designing an appropriate scaffold for CTE is a complex procedure, as it is necessary to create a framework with a specific and replicable architecture [76]. It starts with finding the right biomaterial and follows with the right construction technique to prepare an ideal cellular scaffold to cover all the required characteristics. The development of the cartilage-like scaffold must be based on biocompatible and biodegradable biomaterials. At the same time the construction technique must allow the design of various shapes and sizes with a controlled microstructure to promote cellular adhesion, migration, and growth of the cells present in the target tissue. Finally, the stimulation of cartilage phenotype, the production of cartilage-specific ECM (e.g., aggrecan and collagen type 2), and the maintenance of the desired cellular morphology are essential for scaffold use in CTE [77]. Failure to provide the mentioned characteristics leads to cell dedifferentiation and altered gene expression from cartilage-specific to (most commonly) fibroblastic type [78], a complex challenge that so far has not been conquered.

Additive manufacturing, e.g., 3D bioprinting, is often used to tackle this challenge by the automated fabrication of components in an LbL manner with precisely controlled deposition of a combination of biomaterials, cells, and bioactive molecules, collectively known as bioink [79]. However, the scaffold’s fabrication using 3D bioprinting requires a specific set of properties of the bioink to create an appropriate 3D-bioprinted structure for a specific application, e.g., in CTE to simulate the structure and function of native cartilage tissue [80]. Many attempts have been made to create an ideal bioink for CTE by considering the proper mechanical, rheological, and biological properties of native cartilage tissue [81,82,83]. It is essential to ensure the correct functionality, cell function, and regeneration of the bioprinted tissue. Otherwise, the progress of tissue regeneration and its translation into clinical practice is restricted [84]. The ultimate goal in CTE is to create artificial 3D tissue formulations that can be used in in vivo applications [85,86]. Therefore, we suggest the screening of selected biomaterials, their combinations, and appropriate protein adsorption and surface cellular reactions, to assure appropriate conditions before 3D printing. To avoid high costs and time-consuming experiments, fast, reliable, highly sensitive, and low-cost methods are high in demand to test the suitability of biomaterials for CTE scaffolds.

In this context, QCM can be effectively combined with LbL thin-film preparation (QCM–LbL) [87] to systematically define the components for a specific TE bioink formulation. For instance, appropriate combinations of biomaterials are firstly selected, and cellular behavior is further examined, as shown in Figure 1. In such a manner, the whole process of artificial tissue formation can be divided into several independent steps that can be monitored in real-time to create the final 3D-bioprinted product. These include the choice of biomaterials, their interactions at nanoscale level, initial cellular behavior, and cell–biomaterial interactions [88,89]. The possibility to detect any deviations from the desired goal in the intermediate stages of 3D tissue formation enables rapid optimization and adjustment of several crucial parameters to assure suitable conditions for tissue growth and favorable outcome [90,91].

Recent advances in terms of stability, sensitivity, availability, and high throughput capacity make QCM highly applicable for investigating complex biological problems in advanced research areas such as nanomedicine and TE [92].

Some of the applications of QCM-D in biomaterials studies include interaction studies (i.e., mass and thickness measurement, the kinetics of reaction) during the formation of ultrathin single [93] or multilayers [94,95] of biomaterials in liquid phase on various solid surfaces, solid–liquid interactions of biological molecules such as proteins [96,97], cells [98], and microorganisms (e.g., antifouling) [31] with various functional solids, enzymatic degradation of biomaterials [99], structural rearrangement of polymers [100], and many others. Particularly, its usefulness in studying interfacial phenomena between biomaterials and cells makes it a fascinating tool for application in TE [101].

In a typical process, thin nanofilms of the studied biomaterials and their combinations are prepared on QCM crystals by selecting one of the numerous available methods such as spin coating, dip coating, self-assembly, flame synthesis, LbL, and others. Using nanofilms reduces material costs in the initial stage of surface interaction testing. Their ease of fabrication makes them suitable for a quick survey of a larger number of biomaterials [102]. This is of the utmost importance in developing cellular scaffolds, especially when using natural materials or ECM components, for example, in CTE [13,23]. It is particularly useful when protein (e.g., ECM components) adsorption to a base biomaterial is intended [87]. This is a common strategy in designing the scaffold for CTE [81,103] since it is well-known that the protein adsorption and morphology of the protein layer on the biomaterial surface govern cellular adhesion, cell function, and activity, such as proliferation, survival, and gene expression [35,104,105]. QCM is also most suitable on the nanoscale, as it becomes very difficult to determine the physical properties of the load on the microscale and impossible on the macroscale. These processes can be monitored with the QCM-D in real time, thereby providing information about protein–biomaterial interactions, hence the functionality of the protein–biomaterial composite [9]. Once a formulation of the protein–biomaterial composite is achieved, the QCM-D can be further applied as a biosensing platform to evaluate interactions of the protein–biomaterial composite with cells [106].

The possibility of monitoring cell biological phenomena in real time is pivotal in the early stage of TE. Therefore, it is important to evaluate whether the surfaces of engineered biomaterials can induce desirable initial cell interactions, as this affects the outcome of artificially engineered tissues [102,107]. Cell adhesion is an essential aspect of cellular behavior since this affects other basic cellular responses, such as growth, migration, and differentiation [108]. Accordingly, innovative methods to study cellular behavior in their native state can advance understanding their response to various biomaterials.

### 4.1. Problem of Material Choice, Nanotechnology, Surface Topography, and the Role of Nanofilms

Developing new biomaterials for biomedical applications has made tremendous progress, and we have several biomaterials at our disposal [109]. There have been many attempts to identify suitable biomaterials for a particular biomedical application. The scaffold design for CTE is mostly based on a combination of biomaterials to achieve specific tissue requirements, including structural, physical, chemical, and biological properties [13,110,111,112]. The selection of suitable biomaterials for a scaffold design is a key factor since it drives the cellular responses, guides the growth, and mimics the extracellular matrix (ECM) of native tissues [113]. Nowadays, CTE’s main types of scaffolds are polymeric films, hydrogels, and fibrous scaffolds [81,114]. Nanofiber-based scaffolds are emerging as a versatile alternative for TE and regenerative medicine applications [115]. These structures can mimic the architecture of natural human tissue on the nanometer scale and favor cell adhesion, proliferation, migration, and differentiation [116]. However, hydrogel scaffolds made from natural raw materials have become the most popular in CTE due to their excellent biological properties. Their physiological elasticity, smooth surface, and high water content can better simulate the ECM microenvironment of natural cartilage. In addition, significant progress has been made in the preparation of hydrogels containing nanomaterials to produce nanocomposite hydrogels for the CTE [117].

In addition, the surface properties of scaffolds have been recognized as being of utmost importance due to the direct interface between materials, cells, and other tissue components [23]. Therefore, the crucial part of TE is controlling the surface properties of biomaterials and adjusting cellular behavior, which affects the formation of new tissues [118]. Chemical surface properties such as type, density, or sequence (e.g., amino acid sequence in proteins) of functional groups are pivotal in cell attachment and proliferation. It was shown that some hydrophilic functional groups (hydroxyl, amino) promote chondrocyte attachment, while others (carboxylic) inhibit it on scaffolds from polylactic acid [119]. However, the type of functional group alone does not define attachment. For example, their protein sequence (amino acid sequence) defines the attachment and proliferation on surfaces crucially [120,121]. Surface topography has also been identified as the crucial surface parameter affecting cell behavior by altering protein adsorption. Cells respond to the nanostructured surface by altering adhesion, organizing the cytoskeleton, and expressing the desired phenotype [35]. The basis of these cellular responses represents focal adhesions (FA), a specific gateway for a particular cellular type called “nanoimprinting” [122]. It is well-known that the surface properties of every biomaterial influence the adsorption of proteins, which in turn affects integrin-mediated cell adhesion. The surface nanoarchitecture may affect mechanotransduction, leading to conformational changes in the cell cytoskeleton and consequently changing the shape of the nucleus and the phenotype [123]. QCM frequency was shown the be highly sensitive to changes in surface topography. Superhydrophobic surfaces with pillar-like microfeatures prepared by lithography showed a high-pillar-height dependence of the measured frequency. At lower heights, the adsorbed liquid caused high “in-phase” frequency shifts as the pillars were mass-loading. At a critical pillar height, however, the elastic effect of the micropillar resulted in the coupled frequency veering to the “out-phase” [30].

At the same time, different signaling cellular mechanisms can affect proliferation, growth, and differentiation in response to surface characteristics. This simply means that the fate of each cell depends on the initial cell–biomaterial contact [124]. Therefore, two key factors must be monitored simultaneously at an early stage in TE: protein adsorption to the base biomaterial and the following cell response to its surface. Figure 3 represents possible successive events on biomaterial surfaces and material–tissue interaction, which can be monitored by QCM. Initially, water molecules and ions form a hydrated layer on the surface of the biomaterial (Figure 3a). Secondly, the hydration shells of the proteins interact with the water molecules on the surface (Figure 3b). These influence the fundamental kinetics and thermodynamics of subsequent protein layer formation, e.g., protein orientation, coverage, denaturation. This information is fundamental to understanding the subsequent interactions with cells. When cells arrive at the surface, they recognize the structures of the protein adlayers for adherence, spread (Figure 3c), and form an interface on the surface (Figure 3d) [125].

Researchers have found that macromolecular concepts alone cannot solve all issues in biomaterial science, so nanoscale materials are gaining more attention in many biomedical applications, including TE [126]. Interesting materials for biomedical applications are nanoscale polymer nanofilms [23]. These offer great promise in biomedicine as they serve as the interface to explore interactions between biological objects (biomolecules or cells) and various materials [22]. Nanofilms are thin material layers ranging from a nanometer to a few micrometers. They represent the boundary where most physicochemical processes take place. The LbL method for nanofilm preparation is most promising in TE research as it offers the possibility to fabricate layered composites of various biomaterials in situ in the QCM device, which monitors the build-up of such composites. LbL films are easy to fabricate and are susceptible to fine control of the physicochemical properties. In general, the LbL technique is used to develop biomaterials with suitable biological and mechanical properties. Such constructs are believed to represent an appropriate cellular microenvironment with suitable pore size, interconnectivity, and biological activity for inducing cell differentiation towards desired phenotypes [127]. Thus, nanofilms are widely used for biomedical applications in orthopedics (71,72) and CTE [128,129] to investigate different types of biomaterials and their interactions with cells.

### 4.2. QCM for Interaction Monitoring with ECM Components in CTE

Monitoring the fabrication procedure during scaffold formation is vital to ensure product quality and artificial tissue development. Various methods, including scanning electron microscopy, mercury and flow porosimetry, gas pycnometry, gas adsorption, and microcomputed tomography, are available to monitor scaffold formation and its interactions with biological components [130]. However, most of these methods provide a limited insight into the scaffold formation, especially time-resolved solid–liquid interactions between specific scaffold components, cells, and the biological environment. A niche area for QCM offers just that—real-time monitoring of three key steps that enable systematic and thoughtful scaffold formation:Formation of nanofilms (e.g., LbL) from biomaterials (important to choose the suitable materials to form desired surfaces with desired properties),Solid–liquid surface interactions of ECM components with nanofilms (e.g., protein absorption) (important to provide insight into concrete interactions of base scaffold materials with ECM components, key for cell growth);Solid–liquid cellular interactions (e.g., cellular adhesion, growth, cytotoxicity, etc.) (important to understand cell growth dynamics on/in the scaffolds).

Monitoring of LbL nanofilms from biomaterials by QCM is well-documented. By simultaneous frequency and dissipation recording, QCM allows for the identification of several factors influencing LbL build-up, such as electrostatic charges, salt concentration, polymer conformation, and others. QCM proved to be a suitable technique to study the adsorption kinetics during the formation of a multilayer polymer film [131]. The importance of measuring frequency and dissipation changes simultaneously in film-formation processes was well-described in Rodahl’s work where the authors have shown that even very thin (few nm) biofilms dissipate a significant amount of energy. Three main contributors for the high dissipation were identified: a viscoelastic porous structure that is strained during oscillation, trapped liquid moving within the pores or in and out of them, and the load from the bulk liquid increasing the strain [132].

The abovementioned key steps can be performed subsequently or simultaneously. For example, the formation of biomaterial nanofilms is often performed in combination with ECM components. Such an approach lays the groundwork for the use of LbL nanofilms in investigations of biomaterials in CTE [133,134,135]. Degradable biopolymers and ECM components are preferred materials in CTE since they possess outstanding biocompatibility, low immunological response, low cytotoxicity, and excellent capability to promote cell adhesion, proliferation, and regeneration of new tissues. Unlike synthetic materials, ECM-based scaffolds allow for direct attachment of cells because they often possess unique ligands (e.g., amino acid sequences) that bind to specific cell receptors [136]. Naturally derived biomaterials used in CTE may be protein-based (gelatin, collagen, fibroin, etc.), polysaccharide-based (alginate, agarose, chitosan, cellulose, hyaluronic acid (HA), dextran), or made from decellularized tissues [81].

Recently, the novel formulation termed proteosaccharides was introduced, a combination of polysaccharides and various proteins, to mimic the natural cartilage environment and to improve the scaffold’s physiological signaling and mechanical strength [81]. The design of such a composite construct is complex. In addition to the correct choice of biomaterials, it is necessary to study the interactions of the materials to ensure proper protein binding and create a stable construct. It was extensively emphasized that the processes occurring between cells and materials at the nano–bio level govern subsequent cellular behavior and influence tissue formation [137]. In situ monitoring of these phenomena has been found crucial in controlling cell functions. The analysis of interfacial interactions with protein adsorption and initial cell adhesion was well-demonstrated with the QCM-D technique [125].

The LbL technique, in combination with QCM-D analysis, was used to prepare films from collagen (Col1)/chondroitin sulfate (CS) and Col1/Heparin (HN) with mammalian primary chondrocytes. Data generated from the QCM-D observations showed a consistent build-up of films [138]. In addition, functional multilayer scaffolds for in vivo osteochondral tissue engineering were also developed [139].

#### QCM in Measurement of Protein Adsorption on Biomaterials

ECM proteins initially adsorbed on biomaterial surfaces can mediate initial cellular interactions. Therefore, it is of the utmost importance to assess the interaction potential of ECM proteins with scaffold biomaterials (e.g., polysaccharides) and to monitor the formation and stability of the protein fiber growth and adlayers. The simplest method to study such processes is to design thin films based on raw materials meant for the subsequent scaffold formation using the LbL technique. QCM was shown to be useful for measuring the amount of protein adsorbed onto different surfaces and evaluating the rate of fibril growth [130,140,141]. There are several other techniques available to observe fiber growth and protein ensemble, including atomic force microscopy (AFM) [142], total internal reflection fluorescence microscopy (TIRFM) [143], and surface plasmon resonance (SPR) [144] or dynamic light scattering (DLS) [145], etc. However, they mostly do not provide real-time data recording.

Detailed QCM applications for the adlayer monitoring have already been described and reviewed in other journals [125,146]. Protein adsorption is a key factor in cell activity, attachment, surface migration, proliferation, and differentiation towards the desired phenotype. It was reported that the monocomponent protein adsorption and conformational changes could be investigated with the QCM-D technique [125]. For example, the conformation of bovine serum albumin when adsorbed on a silica surface was determined by applying the QCM-adsorbed mass at various pH conditions [147], as shown in Figure 4. Results revealed the highest protein coverage at a pH of around 5, close to the isoelectric point of the protein. The frequency change and the calculated adsorbed mass changed in the following order: pH 3 < pH 4 < pH 4.5 < pH 5 > pH 5.5 > pH 5.7 > pH 6 > pH 9 (Figure 4a). This pointed to a thicker protein layer at pH 5 compared to the layers at pH 3 and pH 9. It was assumed and later confirmed by AFM measurements that the protein binds to the silica surface in a “side-on” fashion at pH5, where the pyramid-shaped protein (proposed by the authors) binds to the surface with the bottom side of the pyramid with the tip exposed outwards, while it binds in a “flat-on” fashion at the other two pH values, resulting in a thinner protein layer.

QCM was also successfully used to study the influence of protein conformation on cell attachment [148]. The authors showed that bovine serum albumin, fibrinogen, and collagen all bind in different conformations on various solid substrates and that certain conformations that minimize the energy after adsorption favor cell attachment [148].

In addition, QCM-D was used to investigate interfacial phenomena between cells and surfaces modified by various serum proteins such as albumin, fibronectin, and collagen with subsequent adsorption of fetal bovine serum (FBS) to form different adlayers [125]. Investigation of the adsorption phenomena enables the control of the structural–chemical properties of nanomaterials. This is possible since various surface properties, such as wettability, free energy, charge, and roughness influence protein adsorption behavior on a solid substrate [149]. However, fibril formation often results in a complex sensor response; therefore, additional microscopic measurements may be used. One of the most recent achievements is the development of a combined QCM-TIRF technique, which allows for the simultaneous measurement of the mass of peptide adsorbed on the sensor surface and the visualization of fibril growth by a TIRF microscope [150,151]. Such comprehensive understanding of the adlayers’ adsorption, viscoelastic properties and surface arrangement is crucial to understanding further interaction of the biomaterials with cells.

### 4.3. QCM in the Measurement of Interactions with Cells

Cell adhesion and detachment are crucial aspects of cell function and biological processes in bioengineering applications, since these parameters affect basic cellular processes such as cellular communication, growth, migration, and differentiation. Among the most important steps in TE is the initial attachment and growth of cells to achieve sufficient biointegration [108]. Recent literature suggests that QCM is a promising method for studying cellular behavior, as it provides data that are pivotal in understanding many biological phenomena in TE. Different aspects of cell adhesion can be evaluated, including the kinetics of cell attachment, spreading, growth, and cytoskeletal changes [152]. The basic use of QCM in cell biology studies is schematically shown in Figure 5 [40]. The shapes of the frequency vs. dissipation graphs can be unique to specific cellular behavior such as cell adhesion to solid surfaces, cytotoxicity, or cytoskeleton modifications. Five stages of cell adhesion can be identified (Figure 5a), including initial adhesion (I), increase in cell–surface attachment points (II) resulting in a homogeneous layer formation, cell spreading (III) with a steady state of spread cells (IV), and production of ECM from the adhered cells (V). An increase in both frequency and dissipation is observed during cell adhesion (I) as a loosely bound hydrated mass of cells attaches to the surface (Figure 2c). The increase in frequency is counterintuitive, as one would expect a decrease during adsorption processes; however, very loosely bound molecules can behave like a coupled oscillator, resulting in a frequency increase [153]. A typical behavior of a decrease in frequency (see Figure 2a) and a continuous increase in dissipation (see Figure 2b) is observed during (II), which identifies the formation of a more homogeneous, thicker, and more viscoelastic cell layer. During cell spreading (III), a mature compact cell layer is formed, the frequency slightly increases as some water molecules are removed, and the dissipation decreases dramatically, as the compact layer is more rigid (see Figure 2b) than the one during cell adhesion. In the final stage of ECM formation, frequency decreases again as more mass is on the surface (see Figure 2a), and a slight decrease in dissipation can be observed as well if the ECM layer is rigid (see Figure 2b). The cytotoxycity of cells can be evaluated as well using the frequency vs. dissipation curve shapes (Figure 5b). In this case, a typical increase in frequency (mass removal) and decrease in dissipation (lower viscoelasticity) is observed during cell detachment (I) from the surface. If the frequency is not changing or decreases slightly in the same scenario (I), water is removed from the cell’s cytoskeleton, indicating cell lysis. This is followed by a frequency increase during cell lysis, as dead cell material is removed from the surface. Cytoskeleton changes (Figure 5c) can be observed in the same way.

Conventional analytical methods cannot follow these initial steps in cell–biomaterial interactions that QCM can monitor, making QCM a tool beyond the current state-of-the-art. In addition, the QCM system is a useful technique for the detection of cell adhesion without the need for detaching cells from the surface or using labeled molecules [73].

The collected data of QCM measurements are unique and require accurate cellular biological phenomena interpretation. However, considering the advantage of the high sensitivity of QCM instruments in detecting subtle changes in cellular activity, it is possible to monitor even tiny changes in the cytoskeleton induced by various environmental conditions. To date, different QCM-based cell biosensors have been applied to monitor the attachment of different cell types, including fibroblasts [154], osteoblasts [106,155], endothelial cells [156], and others. These studies have indicated that the QCM cell biosensor is suitable for evaluating cell attachment in the early stage of TE. QCM successfully detected mesenchymal stem cell (MCS) responses to biomaterial surfaces [157]. Cell adhesion depends on the surface properties of the biomaterial, such as topography, wettability, charge, and protein adlayers. Once cells adhere and spread on the biomaterial surface, various cellular reactions and morphological changes occur. Thus, in situ monitoring of these phenomena on biomaterial surfaces is crucial for controlling cell behavior, function, growth, reproduction, and differentiation [125]. QCM-D measurements provide information about different adhesion processes and interfacial viscoelastic properties depending on the surface material [73]. In addition, the QCM-D technique was also used to detect the rearrangement of the cell cytoskeleton in monitoring cell biological phenomena such as cell migration and differentiation. Furthermore, the ability of QCM-D signals to detect cytoskeletal changes can be exploited to assess cell health or to sense cell death in response to different external conditions [158,159].

### 4.4. Outlook and Prospects of QCM in Tissue Engineering

The main purpose of this review was to highlight the role of QCM in regenerative medicine. QCM is a practical tool for detection of a wide range of molecules and monitoring of bioprocesses in real time due to its unique characteristics such as high sensitivity, low sample quantity, simplicity, low cost, high throughput, and overall versatility [5]. Its practicality and versatility in TE were reflected in works such as the development of QCM-based mammalian cell biosensors [1] or monitoring of nanoscaled solid–liquid interactions in TE such as initial cell binding, focal adhesion formation, cell spreading, and complex changes in the cytoskeleton [152,160,161], or simple monitoring of multilayer formation (e.g., LbL) [87,155].

Looking to the future, the rapid development of biomaterials requires proper selection and preparation before using them as a potential cellular scaffold in TE. Moreover, the clinical translation of bioactive technologies for medical use should be facilitated; therefore, in situ monitoring and understanding of the interfaces are of great importance for clarifying the nature of biocompatibility [162].

In this context, a step-by-step procedure (Figure 1) was proposed in this review to improve the existing CTE scaffold procedures consisting of selection of appropriate biomaterial and scaffold formation (Figure 1, preprocessing 1) and the subsequent monitoring of cellular responses (Figure 1, preprocessing 2).

In addition, QCM can be coupled with other devices such as microscopes and ellipsometers to obtain more detailed morphological and optical properties of studied platforms. Combining the advantage of the high sensitivity of QCM in detecting nanoscale changes in cell conformation and the power of microscopical visualization, a beyond-state-of-the-art setup can be designed to analyze the cytoskeleton changes induced by different stimuli, e.g., growth factors, cell and gene therapies, immunomodulation, and other external stimuli (electrical, mechanical, or magnetic pulses, etc.). In this way, QCM could be practically applied in CTE to monitor tissue growth and investigate the course of chondrogenesis of different cell types (e.g., precursor cells, MCSs, chondrocytes, etc.). Moreover, the potential of QCM to monitor cellular responses in real time can be used in designing multifunctional and smart scaffolds [163]. Hereby, QCM can act as a biosensor at the cell–tissue interface to detect any subtle changes in the tissue construct and to activate various mechanisms (bioelectric signals for appropriate physiological functions) to preserve desired tissue characteristics in the long term. Essentially, this might allow for the construction of a self-sustaining closed system mimicking native in vivo conditions.

## 5. Conclusions

Recently, the utilization of biomedical nanotechnologies and nanomaterials has surged in the field of CTE. In addition, the importance of the initial cellular response to a particular biomaterial has been increasingly emphasized. It is well-known that failure to provide appropriate initial conditions does not allow for the successful implementation of the 3D construct. Therefore, in this paper, we described a method of a coupled QCM-LbL technique that allows for a systematic approach to design 3D tissue constructs with the ability to monitor the entire course of artificial tissue formation without random selection of biomaterial and unknown cellular response before 3D bioink formulation. In conclusion, we believe that the proposed method will greatly help further research of biomaterials and their use in CTE. Furthermore, this allows for the rapid identification of favorable combinations of biomaterials and excludes those that do not provide an adequate cellular response.

## Figures and Tables

**Figure 1 jfb-13-00159-f001:**
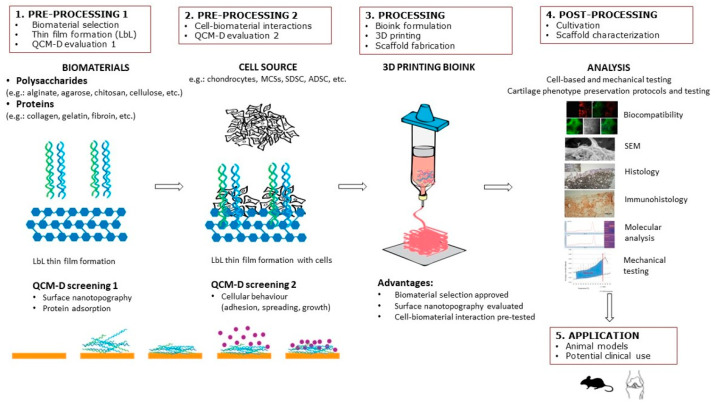
Step-by-step procedure in scaffold design in CTE, including LbL thin-film formation and QCM evaluation.

**Figure 2 jfb-13-00159-f002:**
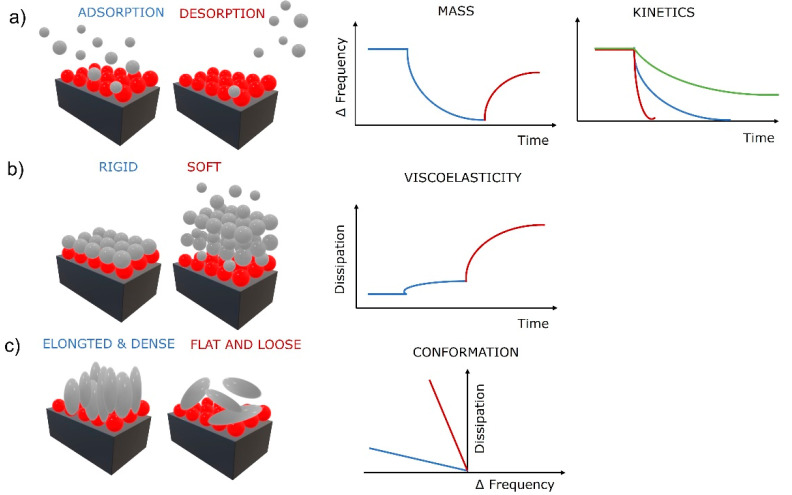
Surface phenomena at solid–liquid interface and the corresponding QCM-D response for (**a**) adsorption and desorption processes, (**b**) evaluation of viscoelasticity, and (**c**) molecular conformationRed balls represent a thin biopolymer film and gray balls, and spheres represent biomolecules (e.g., proteins, cells).

**Figure 3 jfb-13-00159-f003:**
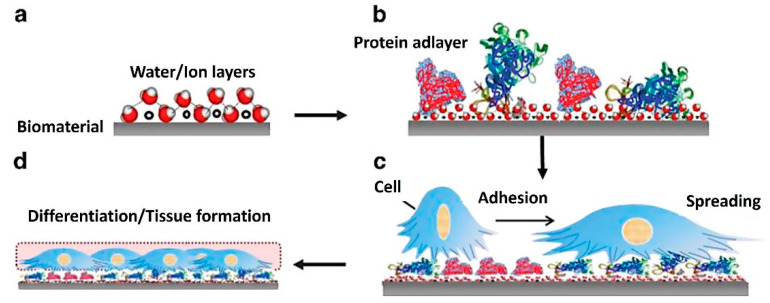
Diagram of successive events on biomaterial surfaces and material–tissue interaction [125]. (**a**) formation of a hydrated layer, (**b**) formation of a protein adlayer, (**c**) cell adhesion and spreading, and (**d**) tissue formation. Reprinted with permission from from [125], SpringerNature.

**Figure 4 jfb-13-00159-f004:**
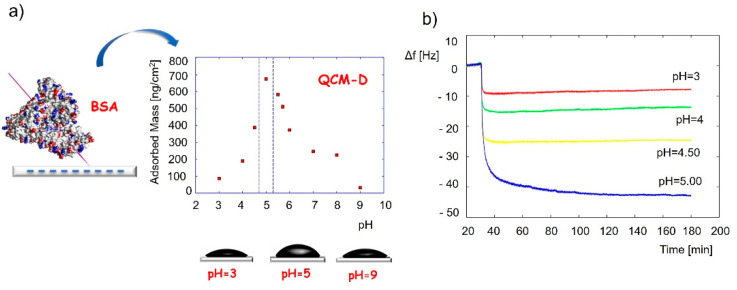
Application of QCM to identify the conformation of adsorbed bovine serum albumin protein. (**a**) Adsorbed mass of the protein vs. the pH at adsorption, and (**b**) QCM frequency changes during adsorption experiments of the protein at different pH values. The frequency change and the calculated adsorbed mass change in the following order: pH3 < pH5 > pH9. Reprinted with permission from [147], Elsevier.

**Figure 5 jfb-13-00159-f005:**
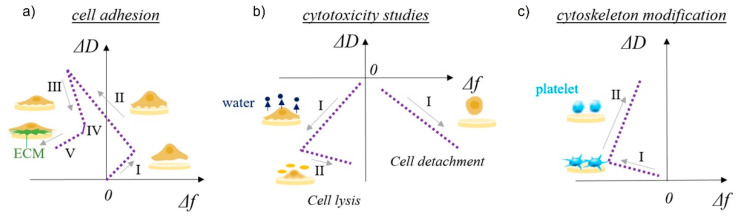
Schematic representation of QCM-D study of multilayer surface characteristics and cell–surface interactions. Δ*D* vs. Δ*f* plots for cell adhesion provide information about (**a**) cell adhesion (I—initial adhesion, II—formation of attachment points, III—cell spreading, IV—steady state of spread cells, V—production of ECM), (**b**) cytotoxicity (left side: I—water release from cytoskeleton, II—cell lysis; right side: I—cell detachment), and (**c**) cytoskeleton modification, such as platelets activation (I—initial interactions among surface sensor and platelets, II—platelets spreading and pseudopodia formation). Reprinted with permission from [73], Tonda-Turo, Carmagnola, and Ciardelli.

## Data Availability

Not applicable.

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
