# Peer review of "Practical Use of Quartz Crystal Microbalance Monitoring in Cartilage Tissue Engineering"

_jfb, 2022, doi:10.3390/jfb13040159_

Round 1
Reviewer 1 Report
This review focuses on the potential use of QCM in CTE. Is there a review similar to this manuscript, and if so, what is the difference?
Why does the article only focus on nanofilms without discussion of other forms of materials?
The cited figures are lack of further explanation. For example, what is the relationship between the cartoon and the curves shown in Figure 2?
It talks LBL technique too much. The manuscript should focus on QCM. Similarly, the reviewer think the manuscript spend much more time on the biomaterial than it should be;
Figure 5, the label on the figures are a and b, but it becomes A and B in the caption.
Author Response
Dear Reviewer,
The replies to your questions, suggestions and other comments are attached.
Thank you
Authors

Reviewer 2 Report
The authors present an interesting review article considering the opportunity of using quartz crystal microbalance-based sensor devices for in-situ evaluation of cell multiplication, differentiation and scaffold formation for the purposes of cartilage tissue engineering. The manuscript is nicely written and relatively well-structure, so it might be redirected for publication if the authors are willing to seriously take into account the following important comments/recommendations:
1) "QCM is a transient mode resonator that measures the frequency of an oscillating quartz crystal, as shown in Figure 2. The oscillating crystal's frequency fundamentally depends on its mass, and mass changes on its interface [30]." - this is a fundamentally wrong statement. In physics, the resonance in solids depends on the particular solid's thickness. When the thickness is half of the wavelength, constructive interference between the incident and reflective wave occurs, so the vibrational amplitude increases. From that point-of-view, the oscillating crystal's frequency does not depend on its mass, but on its thickness. Thinner the crystal, higher the oscillation frequency and in fact, this has been used in QCM technology to increase the sensor sensitivity, but at the expense of higher fragility of the device. The QCM is used as a mass sensor, because when a thin film is added on its surface (the film can be formed by the adherence of proteins, cells or other living matter), the overall crystal's thickness increases, which proportionally increases the wavelength and decreases the resonance frequency. Thus, by knowing the density of the added layer, one can easily convert its thickness into mass per unit area - i.e., the Sauerbrey's equation. Furthermore, the development of QCM technology in 20th and 21st century shows that the sensing mechanism is not restricted only to mass changes, but also to changes in the viscosity-density product of the substance (if it is liquid) or changes in the interfacial wettability as a result of biological binding events. In the latter case, biomolecules with mass below the QCM sensor resolution can be sensed not via their mass loading on the active surface, but rather via their effect on the surface wettability. The reviewer strongly recommends the authors to examine and cite the research of Gunter Sauerbrey, Kanazawa-Gordon, McHale et.al., M. Thompson et.al., Butry et.al., Lucklum et.al., Esmeryan et.al., and perform deeper discussion into the sensing mechanisms of QCM devices. This will not only reveal the versatility of these type of sensors, but might inspire the readers to design QCM biosensors with enhanced sensitivity and resolution towards binding events relevant to tissue engineering. Since this is a review article, the above discussion is mandatory.
Useful links: https://link.springer.com/article/10.1007/BF01337937 ; https://pubs.acs.org/doi/10.1021/ac00285a062 ; https://pubs.acs.org/doi/10.1021/cr00014a006 ; https://www.sciencedirect.com/science/article/pii/S0924424798000065 ; https://ieeexplore.ieee.org/document/4623089 ; https://pubmed.ncbi.nlm.nih.gov/17705513/ ; https://pubs.acs.org/doi/abs/10.1021/ac00025a017 ; https://pubs.acs.org/doi/abs/10.1021/la00027a033 ; https://www.sciencedirect.com/science/article/pii/S0925400516320445 ; https://www.sciencedirect.com/science/article/pii/S0924424720317957 ; https://www.sciencedirect.com/science/article/pii/S0924424718313724 ;- others can be found in the scientific databases.
2) In the Introduction and other places in the text, the authors mention the potential use of QCM sensors for monitoring different cell interactions. It should be mentioned that these devices can be used to evaluate the anti-biofouling performance of different functional coatings and analyze the quality of human semen with direct relevance to artificial insemination.
Useful links: https://www.sciencedirect.com/science/article/pii/S1383586620320529 ; https://pubs.acs.org/doi/abs/10.1021/acsami.0c00353 ; https://www.sciencedirect.com/science/article/pii/S2468233018300367?via%3Dihub ; https://iwaponline.com/jwrd/article/9/1/18/38958/Characterization-of-antibiofouling-behaviors-of ; https://www.sciencedirect.com/science/article/pii/S0264127518307391 ; https://www.sciencedirect.com/science/article/pii/S0924424719308131 ; https://www.mdpi.com/1424-8220/19/1/123
3) Line 177 - the deposition of nanofilms from biomaterials could be implemented on QCM using also the low-cost and scalable flame synthesis technique. The authors are encouraged to discuss this approach.
4) Going deeper into the QCM sensing mechanisms in tissue engineering (showing examples), in light of the current state-of-the-art will enhance the article's quality and will provide insightful opportunities for future work.
5) It is recommended to add a section "Outlook and prospects", where the authors can outline their vision for the future integration of QCM technology in tissue engineering - perspectives, remaining challenges, different designs?
Author Response
Dear Reviewer
The reply to your questions, suggestions and other comments are attached.
Thank you
the Authors

Round 2
Reviewer 1 Report
It is better to incorporate the response to questions 1 and 2 into the manuscript.
Author Response
Dear reviewer,
These are the responses to youe queries:
Q1. It is better to incorporate the response to questions 1 and 2 into the manuscript.
We have incorporated the response into the manuscript. For question 1 in Lines 119-120 and for question 2 in lines 374-385
Thank you,
The authors
Reviewer 2 Report
I cannot recommend further consideration of this paper, because the authors did not take into account (addressed) most of my comments. As the other reviewer commented too, without focusing on the different physical detection mechanisms of QCM, this paper loses its quality.
Author Response
Dear Reviewer,
We are sorry that we didn't efficiently reply to your queries. We think that your review was very comprehensive and helpful. In the latest version of the manuscript we added an additional chapter in the begining of the manuscript (lines 138-224) about the sensing mechanisms of QCM and their practical applications (highlited in yellow) to elucidate QCM basics to the readers more. We agree that it improves the quality of the paper and that the paper would have less impact without it.
Thank you,
The authors
Round 3
Reviewer 2 Report
This version can be accepted for publication.